# Water Jet Erosion Performance of Carbon Fiber and Glass Fiber Reinforced Polymers

**DOI:** 10.3390/polym13172933

**Published:** 2021-08-31

**Authors:** Jesus Cornelio Mendoza Mendoza, Edgar Ernesto Vera Cardenas, Roger Lewis, William Mai, Erika Osiris Avila Davila, Armando Irvin Martínez Pérez, Saul Ledesma Ledesma, Marisa Moreno Rios

**Affiliations:** 1División de Estudios de Posgrado e Investigación, Tecnológico Nacional de México/Instituto Tecnológico de Pachuca, Carretera México-Pachuca Km. 87.5, Colonia Venta Prieta, Pachuca de Soto 42080, Mexico; jesus91cmm@hotmail.com (J.C.M.M.); erika.ad@pachuca.tecnm.mx (E.O.A.D.); armando.mp@pachuca.tecnm.mx (A.I.M.P.); marisa.mr@pachuca.tecnm.mx (M.M.R.); 2Department of Mechanical Engineering, University of Sheffield, Western Bank, Sheffield City Centre, Sheffield S1 3JD, UK; roger.lewis@sheffield.ac.uk (R.L.); wsmai1@sheffield.ac.uk (W.M.); 3Centro de Ingeniería y Desarrollo Industrial, Playa Pie de la Cuesta 702, Desarrollo San Pablo, Querétaro 76270, Mexico; saul.ledesma@cidesi.edu.mx

**Keywords:** water jet erosion, composite materials, vacuum infusion process, wind industry

## Abstract

Complex engineering challenges are revealed in the wind industry; one of them is erosion at the leading edge of wind turbine blades. Water jet erosive wear tests on carbon-fiber reinforced polymer (CFRP) and glass-fiber reinforced polymer (GFRP) were performed in order to determine their resistance at the conditions tested. Vacuum Infusion Process (VIP) was used to obtain the composite materials. Eight layers of bidirectional carbon fabric (0/90°) and nine glass layers of bidirectional glass cloth were used to manufacture the plates. A water injection platform was utilized. The liquid was projected with a pressure of 150 bar on the surface of the specimens through a nozzle. The samples were located at 65 mm from the nozzle at an impact angle of 75°, with an exposure time of 10, 20 and 30 min. SEM and optical microscopy were used to observe the damage on surfaces. A 3D optical profilometer helped to determine the roughness and see the scar profiles. The results showed that the volume loss for glass fiber and carbon fiber were 10 and 19 mm^3^, respectively. This means that the resistance to water jet erosion in uncoated glass fiber was approximately two times lower than uncoated carbon fiber.

## 1. Introduction

Water jet erosion is a wear complex phenomenon that is very difficult to simulate. During decades, many experimental test rigs have been developed in order to study the erosion on different structural materials [1,2]. Currently, wind turbine blades are developed with an optimal strength–weight ratio; therefore, composite materials are widely used for this application [3]. In a previous work, the authors have investigated the effect of erosion in composite materials in order to simulate this kind of wear on the leading edge of wind turbine blades [4]. Experimental tests have led to a useful but incomplete understanding of the phenomenon of raindrop erosion, the effect of different erosion parameters and a general classification of materials based on their ability to resist erosion. However, techniques have not yet been developed to experimentally measure an objective resistance to erosion of materials. Therefore, erosion tests are carried out only to obtain a qualitative assessment of the erosion resistance of materials, as well as to understand their erosive behavior. There are several rain erosion test platforms reported in the literature, in particular rotating platforms, jet erosion platforms, single drop impact platforms and wind tunnel erosion tests [5], in this study the wear will be evaluated using water jet. Sol-gel coatings have been applied to protect against erosion by liquid and solid impact [6]. Contaminant agents accumulate on the blades, generating changes in the surface roughness that alter the flow direction and reduce the efficiency of the wind turbine [7]. Leading edge damage and, therefore, roughness is either caused by subtractive processes such as foreign object damage (bird strikes and debris ingestion) and erosion (hail, rain droplets, sand particles, dust, volcanic ash and cavitation) and additive processes such as filming (from dirt, icing, fouling, insect build-up). These considered applications are focused on wind turbines [8]. Another important aspect is the surface curvature and shape of the water particles, which significantly influence the impact of a high-speed water drop [9]. Erosion due to rain on wind turbine blades is due to repeated impacts of high-speed liquid droplets causing pitting or delamination, reducing the performance of the wind turbine. It has been found that leading edge erosion by rain starts in the zone with a broader curved profile [10]. For modern wind turbines, an increase in the rotor diameter produces high speeds at the tip of the blade, causing rain erosion to become a critical problem [11,12]. To generate significant amounts of power, the turbine must have a large rotor diameter, which results in the fiberglass reinforced polymer blades being up to 100 m long each. When blades of this size are in operation, the tips can travel up to 300 mph. At these speeds, any material is vulnerable to impact; therefore, raindrops can easily damage the blades when they are in operation. The damage created will affect the aerodynamic properties of the blade and, therefore, the power output of the turbine. Despite this problem, wind energy has continued to grow, which is why finding new materials resistant to erosive wear is of great importance to avoid losses in efficiency in the generation of electricity [13,14]. Momber et al. found that the kinetic energy of erosion flow varies due to the changes in the erosion speed and the velocity of the mass flow of erosive liquid particles. The relationship between the volumetric erosion rate and the kinetic energy of the erosive flow has a direct behavior on the power generated [15]. It has been found that the use of coatings applied on the surface of wind turbine blades reduces the maintenance cost against rain erosion [16,17]. The objective of this research work is to carry out an experimental study of water jet erosion on coated and uncoated carbon-fiber reinforced polymer (CFRP) and glass-fiber reinforced polymer (GFRP), in order to simulate the rain erosion caused in the leading edge of wind turbines.

## 2. Experimental Methodology

### 2.1. Specimens

CFRP and GFRP plates were obtained from the Vacuum Infusion Process (VIP), as shown in Figure 1 [18,19]. This process is considered as closed mold. For the manufacture of the carbon fiber sheet, 8 layers of bidirectional carbon fabric (0/90° fabric) were used, and 9 glass layers of bidirectional glass cloth were used for the glass fiber. For both cases, a mixture was made with 304.6 g of epoxy resin Epolam 2015 and 90.9 g of hardener Epolam 2015. Bidirectional fabrics were placed on a previously polished metal plate, in order to obtain a smooth finish. Additionally, a release fabric and an infusion mesh were placed to help unmold the laminate, in this way the resin flows through the fibers that were sealed inside a vacuum bag. Finally, an inlet valve for resin injection and a suction connection for the vacuum pump were installed. At the conclusion of the RI process, 4-millimeter flat sheets of carbon fiber and glass fiber were obtained. Figure 2a,b show the optical microscopy of the surface of the samples of the sheets obtained from carbon fiber and glass fiber, respectively; in both cases the bidirectional tissue is observed (0/90°). In the glass fiber image, the presence of the polymer matrix based on epoxy resin is much more visible. Some samples of carbon fiber and glass fiber were covered with polyester resin (Gelcoat) to determine its behavior during the liquid erosion test. The average thickness of coating was 0.56 mm (Figure 3). This coating has the function of protecting the surface of the composite materials against UV rays. The application of the Gelcoat was carried out with a brush cured at room temperature.

For the measurement of the roughness of the surface of the composite materials, 10 random measurements were made. The average roughness values (Ra) were obtained from an Alicona Infinite Focus 3D optical measurement system (Bruker, Graz, Austria). The hardness of the specimens was determined according to ASTM 2583-95 (ASTM International, West Conshohocken, PA, USA) [20]; 5 measurements per sample were carried out using a Barcol GYZJ934-1 durometer(Barber Colman, Rockford, IL, USA). To observe the surface of the samples, an Alicona Infinite Focus SL electron microscope (Bruker, Graz, Austria) was used.

### 2.2. Test Conditions

The water jet erosion tests were carried out according to ASTM G73-10 (ASTM International, West Conshohocken, PA, USA) [21]. The experimental platform used is shown in Figure 4. The liquid is projected the surface of the flat sheets of composite materials through a nozzle at a pressure of 150 bar using a 25 hp industrial Hydro-pump. The nozzle has a tungsten carbide flat section exit tip with a 110-millimeter long flow stabilizer. The samples to be evaluated are located 65 mm from the end of the nozzle. To determine which of the composite materials has greater resistance to erosive wear, 3 tests were performed on each material at an angle of 75°, with an exposure time of 10, 20 and 30 min.

## 3. Results and Discussion

### 3.1. Roughness and Hardness

Figure 5a shows the acquired values of roughness of all the samples. It can be seen that, due to the manufacturing of the VIP and the operating conditions applied, mean values of roughness, Ra, greater than 1 μm, were obtained. This indicates that the surfaces of the samples have significant irregularities, which influences the wear during the water jet erosion tests [22,23,24]. In addition, it was observed that in the sample of carbon fiber with Gelcoat, there was a wide dispersion of the roughness results. This could be due to the method used to apply the Gelcoat, which, as explained before, was performed using a brush [25]. Figure 5b shows the values obtained for the Barcol hardness. The carbon fiber and glass fiber without Gelcoat show the highest values. This is due to the mechanical properties of the Gelcoat [26], which means that the hardness is lower for samples with Gelcoat [27]. Table 1 shows the average data of roughness and hardness for the materials before the water jet erosion tests.

### 3.2. 3D Optical Microscopy and SEM

Figure 6 shows the scars generated in the composite materials due to the water jet erosion tests, with and without Gelcoat, at an angle of 75° in periods of 10, 20 and 30 min. These test times were established to examine possible failures in the Gelcoat such as cracking, fractures, pitting and loss of adhesion. Once the test started, it was observed that the samples that showed the greatest damage due to the impact of the liquid, in the first instance, are those that contain Gelcoat on the surface, as a result of their low hardness and high roughness [28]. Despite this, Gelcoat can be considered as a coating against water jet erosion, even for a short period. The acquired shape of the scars in the evaluated materials is due to the fact that the nozzle has a tip with a rectangular geometry. At the same conditions, carbon fiber sheet is the material that presented the least damage. This confirms that its performance, in these tests, is due to the good properties of this composite material that were already reported in other studies [29,30]. The evaluated samples did not present damage in the whole area exposed to water jet erosion, only in certain regions and later spread to more vulnerable sections [31].

Figure 7 shows the optical microscopy and the corresponding 3D view of the damaged regions in the evaluated samples of carbon fiber and glass fiber at different times of duration test. It was observed that, after 10 min, the fiberglass presented a deeper wear scar produced by the impact of liquid on the surface. At 20 and 30 min, it was seen that the damage on the surface gradually increased, generating a deeper scar. In comparison, after 10 min, the carbon fiber shows less damage, because the liquid particles only slightly erode the surface, which corresponds to the detachment of the polymer matrix leaving the fibers exposed. After a period of 20 min, small, eroded areas can be observed, and the surface roughness is modified. At 30 min, the repeated impacts of the liquid particles caused the presence of cuts in the upper carbon fibers and the accumulation of removed material around the wear trace. When making a comparison, it is very evident that carbon fiber presented less wear than glass fiber under the conditions in which the liquid erosion test was carried out, this is mainly due to the roughness, the hardness and the type of fabric of each sample [32,33]. The studies carried out have found results where glass fiber has the lowest resistance to wear [34]. Carbon fiber composites can be used in place of glass fiber composites in aerospace applications, because carbon fiber has greater resistance to erosive wear [35].

Figure 8a shows the microscopy of the wear zone corresponding to carbon fiber, in which a slight depth was observed, caused by the number of broken fibers due to the repeated impacts of the liquid on the surface during the test [36]. Figure 8b shows the wear zone of glass fiber where the removal of the polymer matrix and fibers cut randomly by the impact of the liquid on the surface can be seen; in addition, there was the formation of ridges in the direction of impact, which caused an increase in the surface roughness [37,38]. In the sample of carbon fiber with Gelcoat, shown in Figure 8c, the damaged region was observed, where there was the presence of a small remainder of Gelcoat applied on the surface; additionally, small pitting, formed by the continuous attack, was created, which subsequently caused cracks. This leads to the removal of the polymer matrix and the presence of cutting action on the fibers [39,40]. In the case of the glass fiber with Gelcoat, corresponding to Figure 8d, it was observed that the coating was removed quickly at the beginning of the test, leaving the exposed surface of the fiberglass, allowing the subsequent detachment of the polymer matrix and fibers, as well as the formation of ridges, cut fibers and pitting. Once surface damage begins, erosion accelerates due to the roughness changes over the test region of the sample [41,42,43,44]. The impact of the liquid particles on the surface produces gradual wear, modifying its roughness. As the test progresses, the roughness increases, creating valleys and ridges in the impact zone. When a liquid particle collides with this rough surface, the particle tends to lose speed while sliding on the surface, and at the same time, due to the impact of the liquid with a valley or a ridge, the detachment occurs, causing progressive erosive wear.

Figure 9a shows, in greater detail, the damage of the cutting action on the fibers and the removal of the epoxy resin due to the constant impact of the pressurized water on the surface of the carbon fiber sample. In Figure 9b, corresponding to the glass fiber sample, cuts in the fibers and the removal of the epoxy resin were also observed, but in this case, there are some regions that showed greater damage and depth due to the impact of water. This confirms that the resistance to water erosion in glass fiber is lower than in carbon fiber.

### 3.3. Volume Loss and Profilometry

The ASTM G73-10 [21] standard for water jet erosion tests determines that erosion must be reported as volume or mass lost with respect to time. The lost volume was determined from the measurement of the wear scars of samples using a 3D optical microscope. Figure 10 shows the volume lost in the coated and uncoated composites materials. It is observed that the carbon fiber presented, at the end of the duration test, the smallest volume loss; this is due to its good mechanical properties [45,46]. On the other hand, the samples of carbon fiber and glass fiber with Gelcoat are the ones that lost more material during the test, confirming again its low resistance to water erosion [47,48]. It was observed that the behavior between the volume loss and time was linear (Figure 10), due to the progressive damage of the surface and that the increase in roughness is proportional to the increase in erosion wear.

Wear profiles across the scars were obtained. Figure 11a,b show the profilometry of the samples at 10 and 20 min of duration test, respectively. In all cases, it can be observed that at 10 min, the impacts of the liquid particles modified the surface roughness of the materials. In both times, the sample of glass fiber with Gelcoat is the one that presented the scar with a greater depth, while the carbon fiber presented the lowest depth. Figure 11c, corresponding to the exposure time of 30 min, shows that the profiles with high magnitudes of depth were obtained in the samples with Gelcoat and again the sample of carbon fiber is the one that presented the lowest depth in the wear scar during the test, confirming its resistance to water erosion at the conditions tested in this experimental study. Additionally, in the case of coated samples, Figure 11 shows, with dotted lines, the thickness of the Gelcoat in order to observe the depth of wear produced both in the Gelcoat and in the composite material, depending on the test time.

## 4. Conclusions

This investigation was carried out in order to study the performance of coated and uncoated carbon-fiber reinforced polymer (CFRP) and glass-fiber reinforced polymer (GFRP) under water jet erosion, concluding the following reflections.

According to the SEM micrographs, on the eroded zones, it was possible to confirm the presence of cutting action on the fibers, a detachment of the coating and the formation of ridges in the direction of impact, which caused an increase in the surface roughness.From optical microscopy, it was observed that the fiberglass presented a deeper wear scar compared to the carbon fiber where less damage was observed, which corresponds to the detachment of the polymer matrix, leaving the fibers exposed. After 30 min of testing, the repeated impacts of the liquid particles caused the presence of cuts in the upper carbon fibers and the accumulation of removed material around the wear trace.Under the conditions tested in this research work, the resistance to water jet erosion in glass fiber was lower than in carbon fiber. This is due to the good properties of CFRP such as high stiffness, high tensile strength and high modulus, as well as the excellent interaction between the epoxy matrix and fibers.Based on the data obtained, in most of the tests carried out, a linear behavior was observed between the lost volume and the test time, confirming the existence of progressive damage on the surface and concluding that the increase in roughness is proportional to the increase in erosion wear.The water jet erosion platform developed for this research work showed a very acceptable performance, applying a constant pressure throughout the tests and generating a uniform wear on the surface of the tested composite materials.

## Figures and Tables

**Figure 1 polymers-13-02933-f001:**
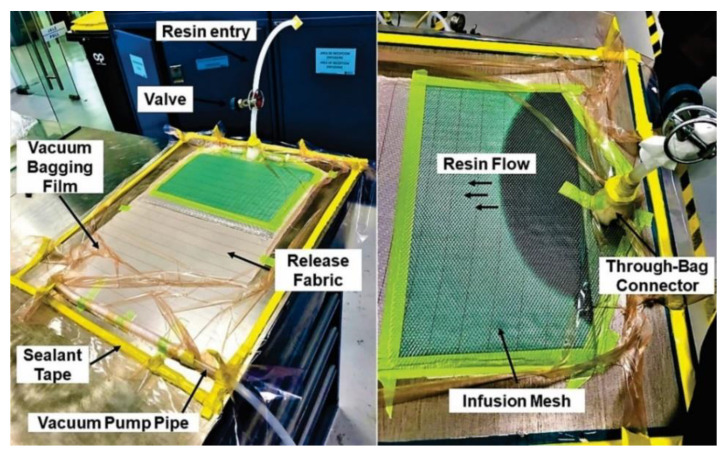
RI process for the manufacture of composite materials.

**Figure 2 polymers-13-02933-f002:**
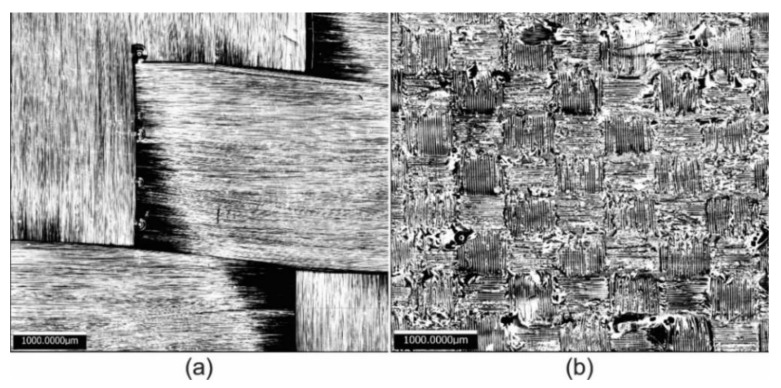
Optical microscopy of (**a**) carbon fiber and (**b**) glass fiber sheets.

**Figure 3 polymers-13-02933-f003:**
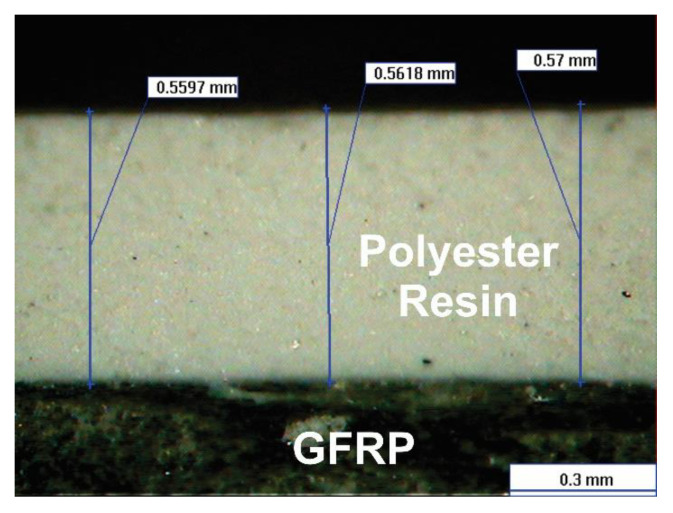
Cross section of glass fiber reinforced polymer and coating of polyester resin.

**Figure 4 polymers-13-02933-f004:**
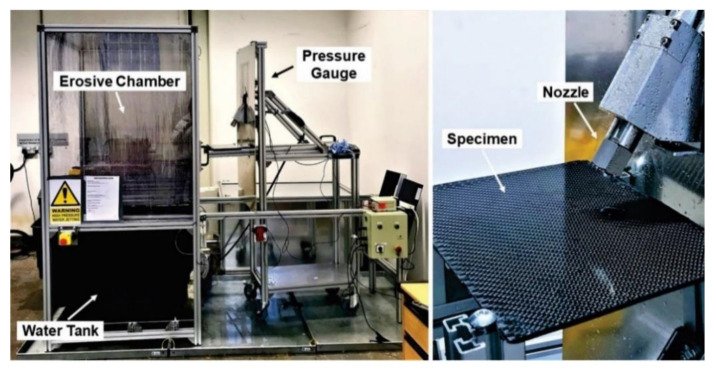
Water jet erosion platform.

**Figure 5 polymers-13-02933-f005:**
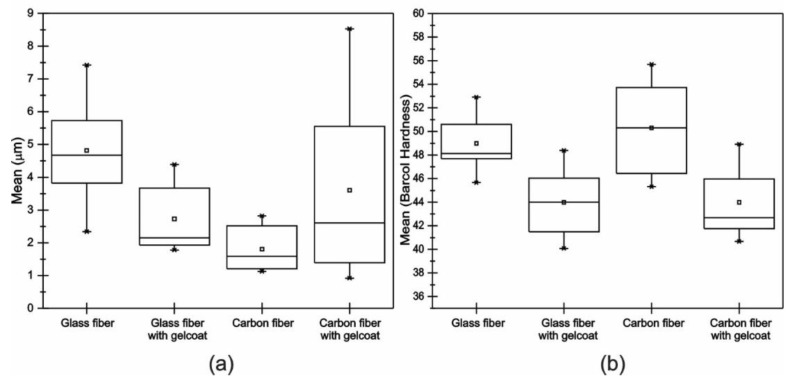
Box diagram of (**a**) roughness and (**b**) hardness.

**Figure 6 polymers-13-02933-f006:**
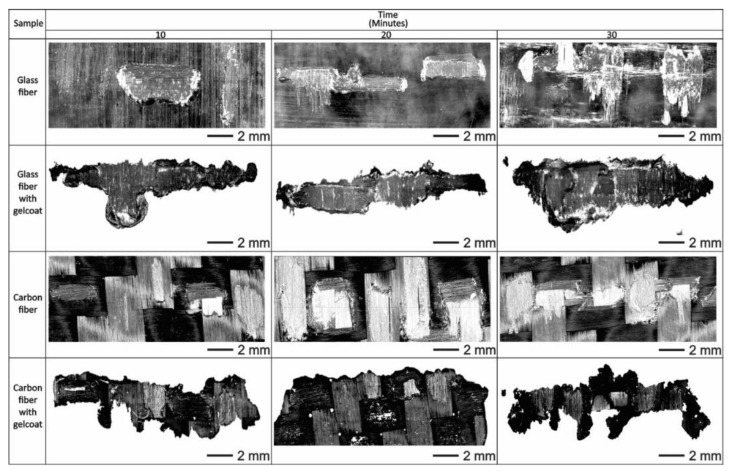
Damage of the surface of the samples using the test of liquid erosion at an impact angle of 75°.

**Figure 7 polymers-13-02933-f007:**
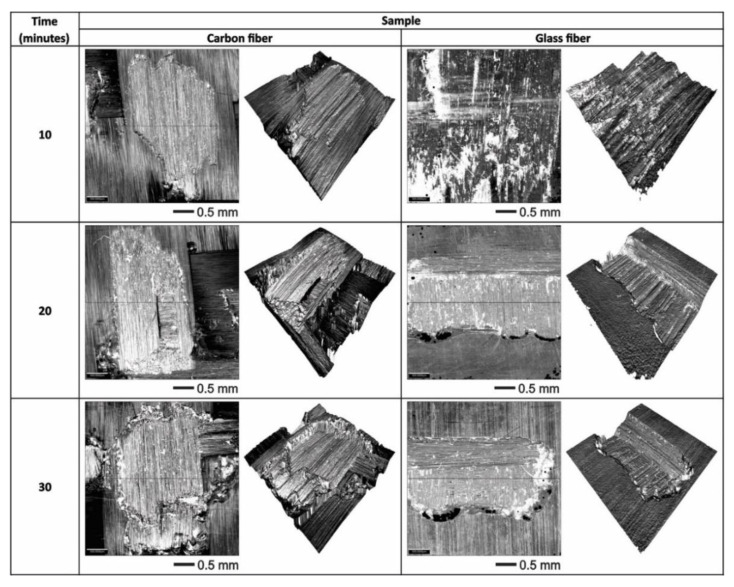
Optical microscopy and 3D view of the wear scars of the samples evaluated using water jet erosion.

**Figure 8 polymers-13-02933-f008:**
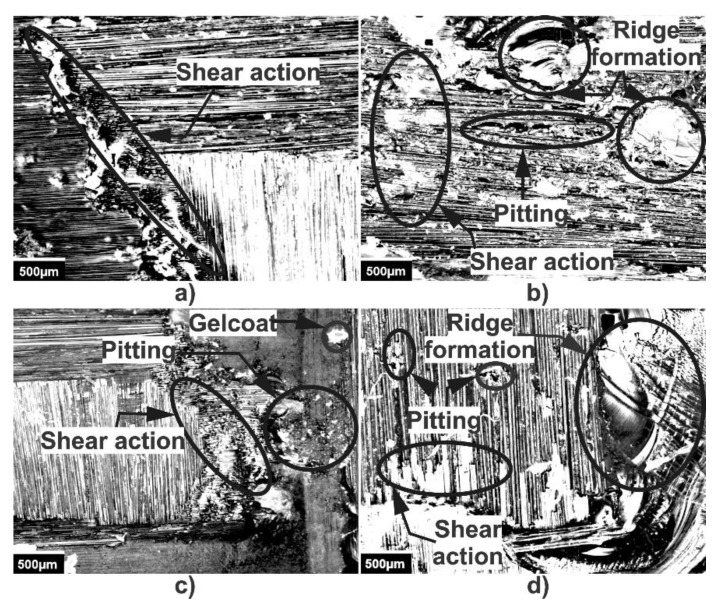
Wear mechanisms by water erosion in (**a**) carbon fiber, (**b**) glass fiber, (**c**) carbon fiber with Gelcoat and (**d**) glass fiber with Gelcoat.

**Figure 9 polymers-13-02933-f009:**
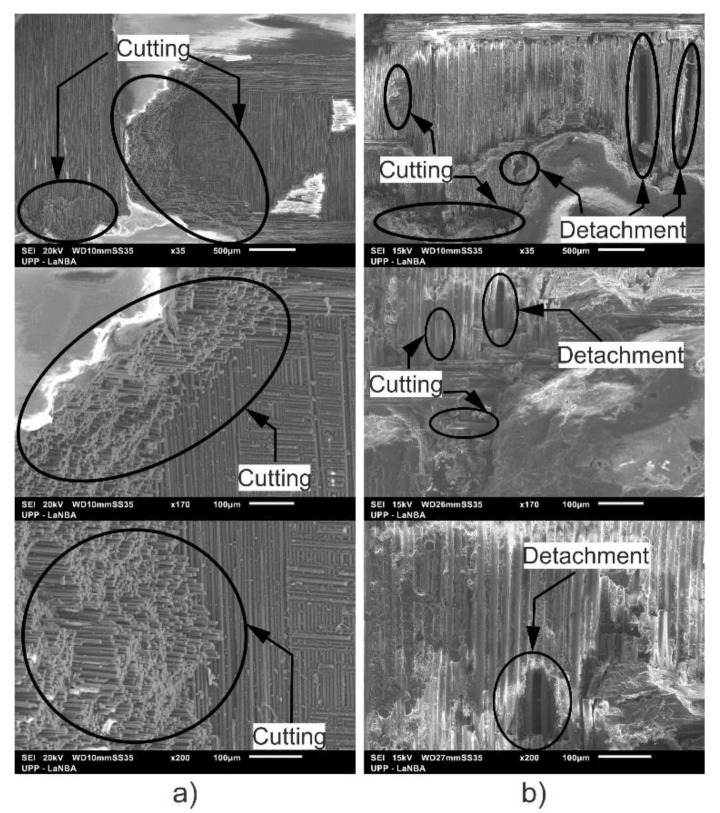
Cutting and detachment action: (**a**) carbon fiber and (**b**) glass fiber.

**Figure 10 polymers-13-02933-f010:**
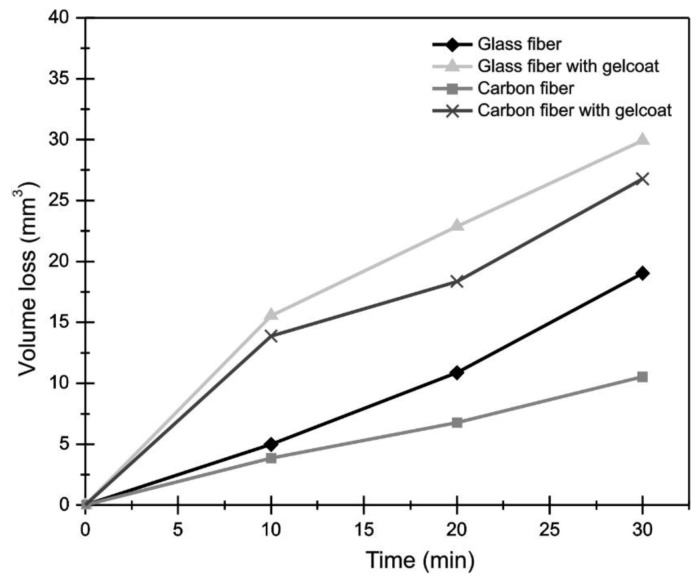
Volume loss during the liquid erosion test with 75° impact angle.

**Figure 11 polymers-13-02933-f011:**
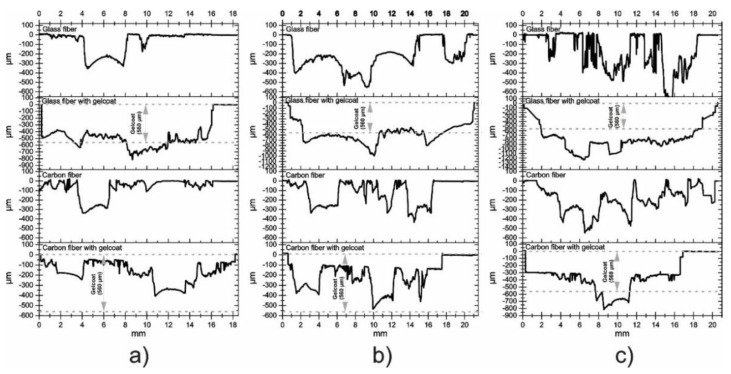
Profilometry of carbon fiber and glass fiber samples with and without Gelcoat with a duration test of (**a**) 10 min, (**b**) 20 min and (**c**) 30 min.

**Table 1 polymers-13-02933-t001:** Average values of roughness and hardness of the materials tested.

Material	Roughness Ra, μm	Hardness Barcol
Glass fiber	4.484	49
Glass fiber with Gelcoat	2.731	44
Carbon fiber	2.074	50
Carbon fiber with Gelcoat	3.610	44

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
