# Peer review of "Water Jet Erosion Performance of Carbon Fiber and Glass Fiber Reinforced Polymers"

_polymers, 2021, doi:10.3390/polym13172933_

Round 1

Reviewer 1 Report

This manuscript is interesting and can be published after minor revision. The following comments:

(i) A "Crosscheck" analysis of your paper indicates significant similarity with other published scientific papers.  I have looked at this carefully.  In many cases, whole paragraphs or significant word groupings were copied without appropriate citations.  Please carefully review and edit this paper to be sure you are fully attributing your sources. This applies to previously published work of yourself or co-authors which must also be cited.  Failure to do so can result in significant copyright issues and possibly charges of plagiarism.

(ii) Elaboration needs more in the introduction section by providing a literature review. 

(iii) What is the novelty of the work?

(iv) Why the authors used SEM and optical microscopy to observe the damage on surfaces?

(v) Could you please compare your findings with the literature about the resistance to water jet erosion in glass fiber was lower than in carbon fiber. 

(vi) In conclusion, the authors need to rewrite the exact findings of optical microscopy from results and discussion. 

(vii) English needs to be correct throughout the manuscript).     

Author Response

RESPONSE TO REFEREE COMMENTS

Authors would like to thank the editor and reviewers for their contributions to the review of the paper.

REVIEWER 1

Comments and Suggestions for Authors

This manuscript is interesting and can be published after minor revision. The following comments:

(i) A "Crosscheck" analysis of your paper indicates significant similarity with other published scientific papers.  I have looked at this carefully.  In many cases, whole paragraphs or significant word groupings were copied without appropriate citations.  Please carefully review and edit this paper to be sure you are fully attributing your sources. This applies to previously published work of yourself or co-authors which must also be cited.  Failure to do so can result in significant copyright issues and possibly charges of plagiarism.

R: The previously published work by the authors have been cited in the Introduction section. A Crosschek analysis was made using the Turnitin tool in order to verify a significant similarity and correct the paragraphs if necessary. 

(ii) Elaboration needs more in the introduction section by providing a literature review. 

R: Some new references were added in order to reinforce the Introduction section. References [4], [5], [6], [9], [14], [15].

(iii) What is the novelty of the work?

R: There are no similar works in the literature on the study of erosive wear by water jet in fiberglass and carbon fiber composites using the platform described for wind turbine blade applications. The results of this study could be used to make decisions in the process of design and selection of materials in the manufacture of wind turbine blades. As well as in the possible application of coatings that reduce erosive wear in this type of elements.

(iv) Why the authors used SEM and optical microscopy to observe the damage on surfaces?

R: In this work, it was decided to use three-dimensional optical microscopy (Figure 7) to see the size and shape of the wear zone in better detail. Likewise, this visualization tool used offers a better method to calculate the wear volume, necessary to know the resistance to erosive wear produced by water jet.

On the other hand, the use of SEM (Figures 8 and 9) allows to know with greater precision the present wear mechanisms, such as microcracks, pitting, grooves, etc., which cannot be identified in optical microscopy.

(v) Could you please compare your findings with the literature about the resistance to water jet erosion in glass fiber was lower than in carbon fiber. 

R: The findings about resistance to water jet erosion of glass fiber and carbon fiber were compared with some literature (References [35] and [36]).

(vi) In conclusion, the authors need to rewrite the exact findings of optical microscopy from results and discussion.

R: A new conclusion was included according to the reviewer´s recommendation.

(vii) English needs to be correct throughout the manuscript).

R: English was revised throughout the document. The manuscript was checked with the support of native English people, correcting some grammatical errors.

Reviewer 2 Report

 The present study explores an experimental study of water jet erosion on coated and uncoated carbon-fiber-reinforced polymer (CFRP) and glass-fiber-reinforced polymer (GFRP) to simulate the rain erosion caused in the leading edge of wind turbines.

The paper was well written, however with minor corrections.

The abstract

It should be to authors to state the findings from their results in this section in qualitative terms.

Fig 8, how the authors need to improve on the resolution of the images at fig 8(a-d). The description of the features on each image can be improved upon.

Under Experimental methodology

Nothing was said about hardness and wear, yet results were presented on these two. I assume it was an oversight.

The equation used to compute the volume loss should be stated in the manuscript as it would be of interest to readers.

It should be to authors to identify the features in the SEM images described in Fig. 9. This can be done by labeling/inserting text in relevant phases in the SEM images.

Author Response

RESPONSE TO REFEREE COMMENTS

Authors would like to thank the editor and reviewers for their contributions to the review of the paper.

REVIEWER 2

Comments and Suggestions for Authors

The present study explores an experimental study of water jet erosion on coated and uncoated carbon-fiber-reinforced polymer (CFRP) and glass-fiber-reinforced polymer (GFRP) to simulate the rain erosion caused in the leading edge of wind turbines.

The paper was well written, however with minor corrections.

The abstract

It should be to authors to state the findings from their results in this section in qualitative terms.

R: Some findings from results were added in the abstract in qualitative and quantitative terms.

Fig 8, how the authors need to improve on the resolution of the images at fig 8(a-d). The description of the features on each image can be improved upon.

R: The resolution of Figure 8 was improved, as well as labels and signs.

Under Experimental methodology

Nothing was said about hardness and wear, yet results were presented on these two. I assume it was an oversight.

R: Effectively, separate results are given for both, hardness (Figure 5 and Table 1) and wear volume (Figure 10). A little discussion about the relationship between these two parameters is included in 3.3. Volume loss and profilometry section, we hope this responds to the reviewer's comment, otherwise, please tell us to attend the new observations.

The equation used to compute the volume loss should be stated in the manuscript as it would be of interest to readers.

R: In this study equations were not used to calculate the wear volume. The lost volume was determined from the measurement of the wear scars of samples using an Alicona Infinite Focus 3D optical microscope. This was indicated in the same section.

It should be to authors to identify the features in the SEM images described in Fig. 9. This can be done by labeling/inserting text in relevant phases in the SEM images.

R: Features of wear mechanisms, by labeling/inserting text, were included in Figure 9.
